# Meta-analyses of *Culex* blood-meals indicates strong regional effect on feeding patterns

Jet S. Griep[1]*, Eve Grant[1], Jack Pilgrim[1], Olena Riabinina[2], Matthew Baylis[1], Maya Wardeh[1,3], Marcus S. C. Blagrove[1]*

1 Institute of Infection, Veterinary and Ecological Sciences, University of Liverpool, Liverpool, United Kingdom, 2 Department of Biosciences, Durham University, Durham, United Kingdom, 3 Department of Computer Science, University of Liverpool, Liverpool, United Kingdom

* jet.griep@liverpool.ac.uk (JSG); grte0276@liverpool.ac.uk (MSCB)

**Editor:** Felix Hol, Radboud University Nijmegen Radboud Institute for Molecular Life Sciences: Radboud Universiteit Radboud Institute for Molecular Life Sciences, NETHERLANDS, KINGDOM OF THE

## Abstract

Understanding host utilization by mosquito vectors is essential to assess the risk of vector-borne diseases. Many studies have investigated the feeding patterns of *Culex* mosquitoes by molecular analysis of blood-meals from field collected mosquitoes. However, these individual small-scale studies only provide a limited understanding of the complex host-vector interactions when considered in isolation. Here, we analyze the *Culex* blood-feeding data from 109 publications over the last 15 years to provide a global insight into the feeding patterns of *Culex* mosquitoes, with particular reference to vectors of currently emerging *Culex*-borne viruses such as West Nile and Usutu. Data on 29990 blood-meals from 70 different *Culex* species were extracted from published literature. The percentage of blood-meals on amphibian, avian, human, non-human mammalian, and reptilian hosts was determined for each *Culex* species. Our analysis showed that feeding patterns were not significantly explained by mosquito species-level phylogeny, indicating that external factors play an important role in determining mosquito feeding patterns. For *Cx. quinquefasciatus,* '*Cx. pipiens* pooled', and *Cx. tritaeniorhynchus,* feeding patterns were compared across the world's seven biogeographical realms. *Culex tritaeniorhynchus,* '*Cx. pipiens* pooled' and *Cx. quinquefasciatus* all had significantly varied feeding patterns between realms. These results demonstrate that feeding patterns of *Culex* mosquitoes vary between species but can also vary between geographically distinct populations of the same species, indicating that regional or population-level adaptations are major drivers of host utilization. Ultimately, these findings support the surveillance of vector-borne diseases by specifying which host groups are most likely to be at risk.

## Author summary

Being aware of mosquito biting behaviour is essential to determine the threat of mosquito-borne diseases. Studying the feeding patterns of *Culex* mosquitoes is crucial as these mosquitoes are vectors of currently emerging or re-emerging arboviruses such as West Nile and Usutu. Feeding behaviour of *Culex* mosquitoes has been examined in many individual small-scale studies. These studies only focus on

**Data availability statement:** All relevant data are in the manuscript and its supporting information files.

**Funding:** JSG acknowledges funding from Biotechnology and Biological Sciences Research Council (BBSRC), Newcastle, Liverpool, Durham Doctoral Training Program (NLD DTP). MSCB and MB acknowledge funding from UK Research and Innovation (UKRI)/ Department for Environment, Food and Rural affairs (DEFRA): BB/X018172/1. MSCB and MB also acknowledge funding from Biotechnology and Biological Sciences Research Council (BBSRC)/Department for Environment, Food and Rural affairs (DEFRA): BB/W002906/1. The funders had no role in study design, data collection and analysis, decision to publish, or preparation of the manuscript.

**Competing interests:** The authors have declared that no competing interests exist.

the feeding patterns in a specific area. To gain a more global understanding of these feeding patterns we analyzed all available *Culex* blood-feeding data from the last 15 years. In summary, data on 29990 blood-meals from 70 different *Culex* species was collected. For each species the percentage of blood-meals on different host groups was determined. We analyzed the relationship between feeding patterns and mosquito phylogeny, which showed that phylogeny alone could not explain feeding patterns. These results indicate that external factors such as land use and climate could play an important role in determining feeding patterns. A more in-depth analysis of the feeding patterns for three important vector species, *Cx. quinquefasciatus*, '*Cx. pipiens* pooled', and *Cx. tritaeniorhynchus* was done in different biogeographical realms. All three species showed different feeding patterns in the included realms. Thus, the same species can have different feeding patterns in different regions, indicating the importance of local surveillance.

## Introduction

*Culex* mosquitoes can be vectors for viruses such as Japanese encephalitis virus, Rift valley fever virus, and several others [1–3]. Many *Culex* species can act as vectors, transmitting pathogens between reservoir hosts, or transmitting from reservoir hosts to dead-end hosts which, in some cases, includes humans. Several *Culex* species have been implicated in the transmission of emerging and re-emerging diseases such as West Nile and Usutu [4–6], demonstrating the ongoing risk posed by mosquitoes of this genus.

Assessing the feeding patterns of mosquito vectors is essential in determining the epidemic potential of certain mosquito-borne pathogens. The hosts that mosquitoes have fed on can be determined by molecular analysis of the blood-meals in the midguts of field collected mosquitoes [7]. If there is a high percentage of both human and reservoir blood-meals and an overlap in time and space of the vector and the host, the risk of efficient pathogen transmission to humans increases significantly [8]. As feeding patterns can shift due to both climate and land use change, it is important to be aware of recent feeding patterns [9,10].

Both intrinsic and extrinsic factors can influence feeding patterns. Intrinsic factors such as genetics can affect host preference [11], but do not fully explain feeding patterns. For example, despite having limited genetic differences *Culex pipiens pipiens* and *Culex pipiens molestus* have very distinct feeding patterns. *Culex pip. pipiens* is believed to feed more on birds whilst *Culex pip. molestus* is believed to mostly utilize mammalian hosts. The mechanism underlying this difference in feeding patterns is unknown [12], but is likely influenced by external factors. Examples of these are host availability and seasonality [13]. Host defensive behaviors and learning could also influence host utilization. For example, if a mosquito had a successful feed on a host it is more likely to go for the same host species for the second feed [14]. Since both intrinsic and extrinsic factors can play a role in host selection of mosquitoes the exact mechanism behind this is hard to elucidate.

Many species of the *Culex* genus are able to feed on a wide variety of hosts [15]. *Culex* feeding patterns have been studied frequently in small-scale individual studies, focusing on the regional feeding patterns of the mosquitoes from that area. As feeding patterns are affected by a variety of external factors it is expected that they might also differ between biogeographic realms. These realms represent an area that has a similar plant and animal distribution and that experienced a distinct evolutionary history and climate [16]. Previous analyses have shown that mosquito feeding patterns are associated with major continental drift events [17].

As the realms have different assemblages of animals and plants; but share certain mosquitoes, we wanted to explore feeding patterns of important vector species in these realms. This could show if different realms require specialized surveillance strategies.

Here, we analyze the data published over the last 15 years describing *Culex* blood-feeding in the field. We first look at the whole genus level and then focus in on three important *Culex* vectors. For the whole *Culex* genus we analyze whether there is a pattern between phylogenetic relatedness and feeding patterns. For three important vector species we assess the relationship between biogeographical realm and feeding patterns. Overall, this analysis enables a global insight into the feeding patterns of *Culex* mosquitoes and thus provides higher-level data for risk assessments to specific hosts, and to specific regions, which will inform ongoing efforts, particularly in Europe [18,19], to mitigate the spread of *Culex*-borne viruses (e.g., West Nile virus and Usutu virus) [20].

## Methods

### Data collection

**Meta-analysis.** *Culex* blood-feeding studies were collected using the PubMed database. The combination of keywords: "*Culex*", "Cx." "blood meal", "bloodmeal", "blood-meal", "mosquito", "host-species", "host", and "feeding" was used to find relevant publications. For several species, where data were lacking, additional searches were performed included the full species name. No additional records were found using this strategy. English literature from 2008–2023 was included in this review. This 15-year period was chosen to focus on recent feeding patterns of *Culex* mosquitoes. A total of 288 records were identified for screening. All abstracts were screened and all records that did not describe blood feeding of *Culex* mosquitoes were excluded. For the remaining 153 records the full text was analyzed. Papers were included in our analysis if they specified the number of blood-meals and if they utilized molecular methods to confirm the blood-meal source. Molecular methods that were accepted for inclusion were: PCR, mNGS, ELISA, antiserum, and agarose gel diffusion. Studies performed in zoos were included, as they are a location where mosquitoes are free to choose or not choose for a specific host under field conditions. Review studies were not included. Studies which analyzed the presence of *Culex* species in animal traps and laboratory-based blood-feeding studies were excluded. Additionally, studies which introduced mosquitoes from the lab to the field conditions were excluded as well. The full overview of our screening process can be found in S1 Fig.

From the articles that were included the following information was extracted: PMID, Title, Authors, Publication year, DOI, Mosquito species, Mosquito location (country), Biogeographical realm, Host (to level described in literature), Host (major host group: amphibian, avian, human, non-human mammalian, and reptilian), Number of blood-meals, Total number of blood-meals (in that specific study), Collection method, Blood-meal analysis method, Land use type (of where study was done), Indoor/outdoor collections. The realm was determined based on the Udvardy (1975) system [21]. If the specific form of *Culex pipiens* was not specified, the data was categorized under *Culex pipiens* not specified (ns). Furthermore, for certain parts of our analysis *Culex pipiens* (ns), *Culex pipiens molestus, Culex pipiens pallens, Culex pipiens pipiens, Culex pipiens/molestus* hybrid were clustered as '*Culex pipiens* pooled'. In the dataset they were all kept separate if they were recorded to the biotype level in the original paper. The hosts were described at the level recorded in literature. All data collected for this study and the code used to analyze the data are available in S1 Database and S1 Code. The publications per *Culex* species are cited in S1 Table.

## Data analysis

**Feeding patterns.** The number of blood-meals per host species was recorded. All hosts were aggregated into five main groups: amphibian, avian, human, non-human mammalian, and reptilian. Based on the total number of blood-meals recorded for each species the percentage of blood-meals from each of these five groups was calculated. All calculations and data visualizations were performed in R software v4.2.0 [22], using the *ggplot2* package v3.4.4 [23] and *maps* package v3.4.1.1 [24]. The base layer of the map was made available from Natural earth (http://www.naturalearthdata.com/about/terms-of-use/). Hierarchical clustering of the feeding patterns was done with the *hclust* function from the stats package v3.6.2 in R utilizing the default settings.

**Phylogenetic tree.** Multiple sequence alignments were made of the cytochrome *c* oxidase subunit I (*COI*) gene. This gene is commonly used for mosquito species identification as it is easy to amplify and has enough variation in its sequence to distinguish between different species [25]. The *COI* gene sequence is available for many different *Culex* species, allowing as many *Culex* species as possible to be included in the phylogeny. The sequences were obtained from NCBI (National Centre for Biotechnology Information) and only the *Culex* species for which a *COI* gene was available were included for further analysis. With the available sequences a multiple sequence alignment was made in MEGA11 v11 [26] using the ClustalW algorithm. *Chaoboridae* was included as the outgroup. Gblocks 0.91b was utilized to remove poorly covered regions [27]. Accession numbers for all of the included species can be found in S2 Table. A phylogeny was then created utilizing maximum likelihood methods and 1000 ultrafast bootstraps in IQ-TREE v. 1.6.10 [28].

**Tanglegram.** For the 57 species that had *COI* genes available on NCBI a tanglegram was created which compared *Culex* phylogeny to a tree based on the feeding patterns of the same mosquito. For this tanglegram the Baker's gamma index (BGI) [29] and the Pearson correlation [30] was calculated, which give an estimation of the association of the two dendrograms compared in the tanglegram. To determine the statistical significance of the BGI the labels for one of the trees were shuffled and the BGI was calculated again (n = 999). This tanglegram and the statistical tests were done with the *dendextend* package in R [30].

**Feeding patterns between realms.** The feeding patterns of three major vectors, *Cx. tritaeniorhynchus,* '*Cx. pipiens* pooled', and *Cx. quinquefasciatus* were analyzed across different biogeographical realms. To assess the difference in feeding patterns between realms Pearson's chi-square test was used. We present raw $\chi^2$, raw P vales, and corrected P values using the Bonferroni method in S3 Table, allowing the reader to choose or perform alternate multiple testing should they see fit. This analysis was performed with the stats package (v3.6.2) in R.

**Study characteristics.** For every included *Culex* species the percentage of feeds across different land use types was analyzed. These results are described in S2 Fig. For *Cx. tritaeniorhynchus,* '*Cx. pipiens* pooled', and *Cx. quinquefasciatus* the different collection methods and collection locations (indoor/outdoor) were analyzed as well. These results are presented in S3 and S4 Figs respectively.

## Results

### Spatial distribution of *Culex* blood-feeding publications

In total, 109 publications were identified that fit the criteria of this study. The publications described blood-meals for 70 different *Culex* mosquitoes described. Data on 29990 blood-meals were recorded. The publications contained *Culex* blood-feeding information from 30

different countries. The United States had both the highest number of publications, and the highest number of species described in these publications ([Fig 1]). The correlation between the number of species and the number of publications that was done in each country is quantified in [S5 Fig].

## Comparison between phylogeny and feeding patterns

A total of 57 *Culex* species were included in the comparison between phylogeny and feeding patterns. Based on the clustering of feeding patterns six broad clusters emerged: majority non-human mammal feeds (*Cx. annulioris, Cx. annulirostris, Cx. caudelli, Cx. coronator, Cx. crinicauda, Cx. declarator, Cx. eduardoi, Cx. gelidus, Cx. maxi, Cx. poicilipes, Cx. salinarius, Cx. spissipes, Cx. taeniopus, Cx. theileri, Cx. tritaeniorhynchus*), majority avian feeds (*Cx. australicus, Cx. bitaeniorhynchus, Cx. dolosus, Cx. globocoxitus, Cx. interrogator, Cx. intrincatus, Cx. lactator, Cx. modestus, Cx. mollis, Cx. orbostiensis, Cx. orientalis, Cx. pip. molestus, Cx. pip. pallens, Cx. pip. pipiens, Cx. restuans, Cx. stigmatosoma, Cx. tarsalis*), mix of mammalian and avian feeds (*Cx. erraticus, Cx. erythrothorax, Cx. nigripalus, Cx. perexiguus, Cx. quinquefasciatus, Cx. restrictor, Cx. sitiens, Cx. apicinus*), having amphibian feeds (*Cx. amazonensis, Cx. microculex sp., Cx. territans*), high percentages of human feeds (*Cx. antennatus, Cx. chidestri, Cx. decens, Cx. neavei, Cx. pedroi, Cx. univittatus, Cx. vaxus, Cx. zombaensis*), and majority reptile feeds (*Cx. atratus, Cx. janitor, Cx. melanoconion sp., Cx. pilosus, Cx. bahamensis, Cx. hortensis*). The phylogenetic clustering and the feeding clustering showed a very low correlation (BGI = 0.211, Pearson correlation = 0.203) ([Fig 2]). The number of blood-meals and publications per *Culex* species are found in [S1 Table]. For each species the variation in number of blood-meals on major host groups per study is shown in [S6 Fig]. The same feeding proportions are lined up with the phylogeny, rather than clustered and rearranged, in [S7 Fig].

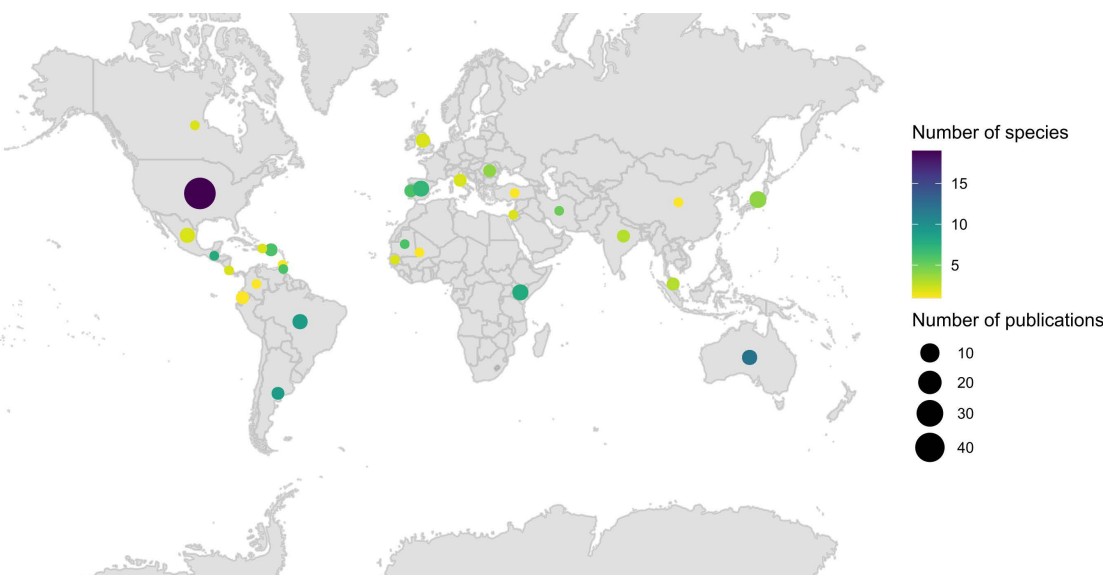

**Fig 1. Spatial distribution of *Culex* blood-feeding publications.** Circle size represents the number of publications per country. Colour of the circle represents the number of *Culex* species that were analyzed per country. The map was created using the *maps* package (v3.4.1.1; License: GPL-2; Becker and Wilks 2022) in *R software* [22]. The base layer of the map is available on https://www.naturalearthdata.com.

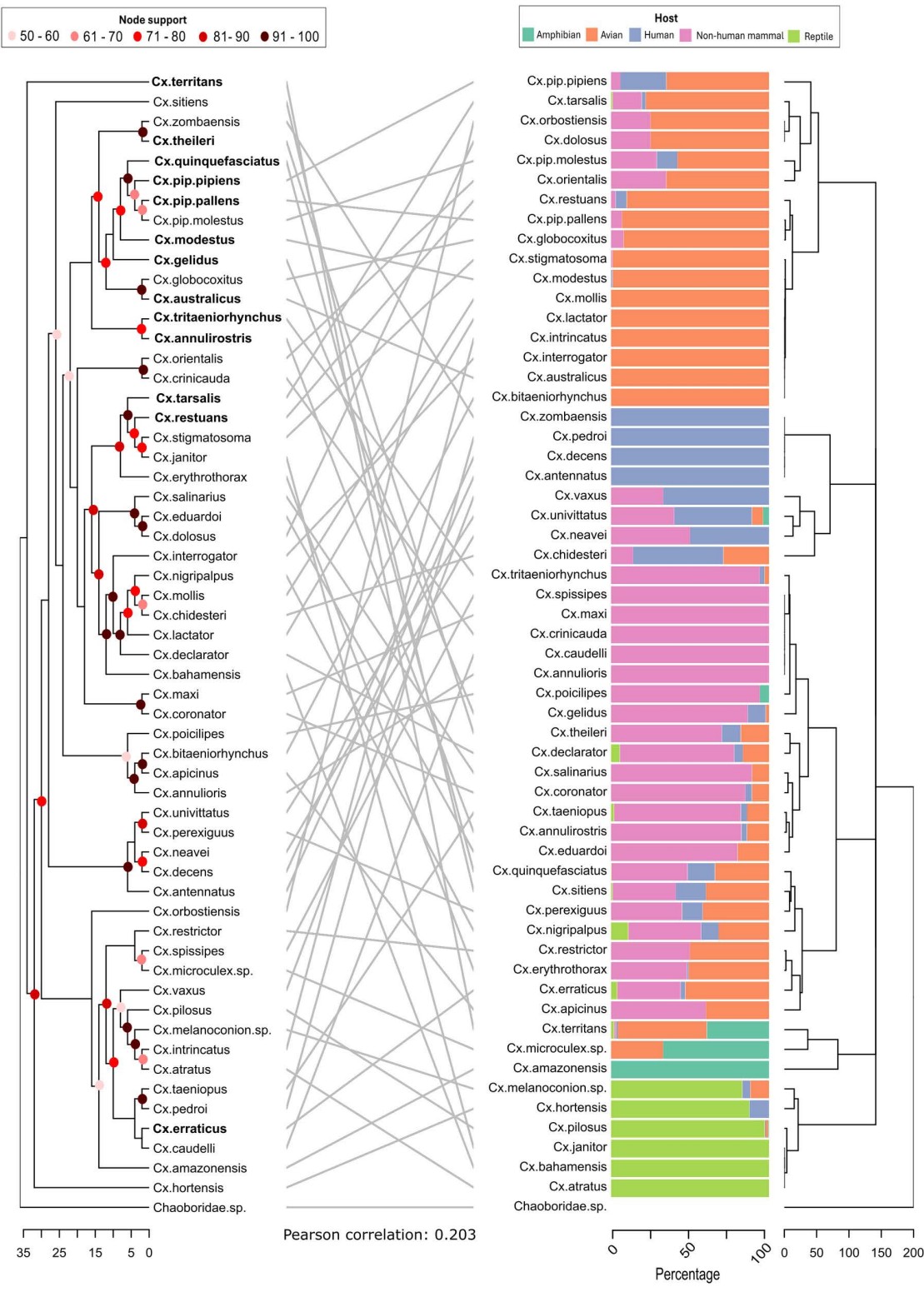

**Fig 2. Tanglegram of a phylogenetic tree based on the cytochrome c oxidase subunit I (COI) gene of 57 *Culex* (*Cx.*) mosquitoes compared to a hierarchical clustering tree of *Culex* mosquito feeding patterns.** The phylogenetic tree (shown on the left) was generated using IQ-TREE with 1000 ultrafast bootstrap alignments. Node support (bootstrap values) is indicated using colored circles in the phylogenetic tree. If the node support was less than 50, no circle was placed on the clade. The labels of species that are known vectors for several medically relevant pathogens are bold in the phylogenetic tree [31]. The hierarchical clustering

tree (shown on the right) was created in R using the *hclust* function. The phylogenetic and feeding tree are connected between the same species. The branch length of the trees represents the distance between clusters. No clusters were found that were the same in both trees. Next to the feeding dendrogram the percentage of feeds on each host group is presented per *Culex* species. The *Chaoboridae* family was chosen as the outgroup. The number of blood-meals and publications per *Culex* species are found in S1 Table. For each species the variation in number of blood-meals on major host groups per study is shown in S6 Fig. The same feeding proportions are lined up with the phylogeny, rather than clustered and rearranged, in S7 Fig.

## Feeding patterns across biogeographical realms

'*Culex pipiens* pooled', *Cx. quinquefasciatus, and Cx. tritaeniorhynchus* were chosen for further analysis by realm as they are important vector species which had feeding pattern available in three or more realms. '*Culex pipiens* pooled' had a majority of avian feeds in the Nearctic, Palearctic and Australasian realms. However, for the Afrotropical realm nearly 90% of '*Culex pipiens* pooled' feeds was recorded on non-human mammals and humans. In the Neotropical realm close to 70% of all blood-meals was recorded on non-human mammals. All of the realms differed significantly in their feeding patterns for '*Culex pipiens* pooled' except for the Nearctic and Palearctic realm ($\chi 2 = 7.59$, df = 3, p > 0.05) (Fig 3). *Culex quinquefasciatus* had a majority of mammal feeds in the Afrotropical realm, human feeds in the Indomalayan realm, and avian feeds in the Australasia and Neotropical realms. In the Nearctic realm there was a similar percentage of avian and mammalian feeding. All realms had significantly different feeding patterns for *Culex quinquefasciatus* except for the Nearctic and Neotropical realm ($\chi 2 = 5.772$, df = 3, p > 0.05) (Fig 3). *Culex tritaeniorhynchus* feeding patterns in the Afrotropical and Indomalayan realms, had a majority of feeds being on non-human mammalian hosts. In the Palearctic realm a majority of avian feeds was observed. All realms had significantly different feeding patterns (Fig 3). In Fig 3 we present our statistical analysis without multiple testing correction. We included the most commonly used and stringent multiple testing correction method (the Bonferroni method) in the detailed analysis in S3 Table. We show that the Bonferroni universal null hypothesis can be rejected, giving further support to our results, and we present raw $\chi 2$, raw P vales, and corrected P values in S3 Table, allowing the reader to choose or perform alternate multiple testing should they see fit.

## Discussion

Our analysis of *Culex* feeding patterns revealed that *Culex* mosquitoes have a wide variety of feeding patterns, with some feeding on only mammals, only birds, or only reptiles, but most of them also feeding on multiple host groups. We show that the feeding patterns do not appear to be explained by phylogeny alone. Furthermore, in depth analysis on the highly sampled, '*Culex pipiens* pooled', *Culex quinquefasciatus, and Culex tritaeniorhynchus* in different biogeographical realms showed significantly different feeding patterns. Assessing the feeding patterns of these mosquitoes is essential as they are primary vectors for several important mosquito-borne viruses, including zoonotic viruses [1,2].

### Overall importance of blood-meal analyses

*Culex* mosquitoes with a wide host range utilization can potentially act as bridge vectors, e.g., for West Nile virus (WNV), transmitting virus from reservoir birds to other susceptible-but-dead-end hosts such as humans and horses. For example, the competent WNV vectors *Culex restuans, Cx. pipiens* (ns)*, Cx. theileri, Cx. quinquefasciatus,* and *Cx. modestus* all had feeding patterns that included both human and avian bloodfeeding [32–35]. Other studies have described *Cx. restuans* and *Cx. pipiens* (ns). to be largely ornithophilic [36], which is in line with our results, with a majority of *Cx. restuans* and *Cx. pipiens* (ns) blood-meals being

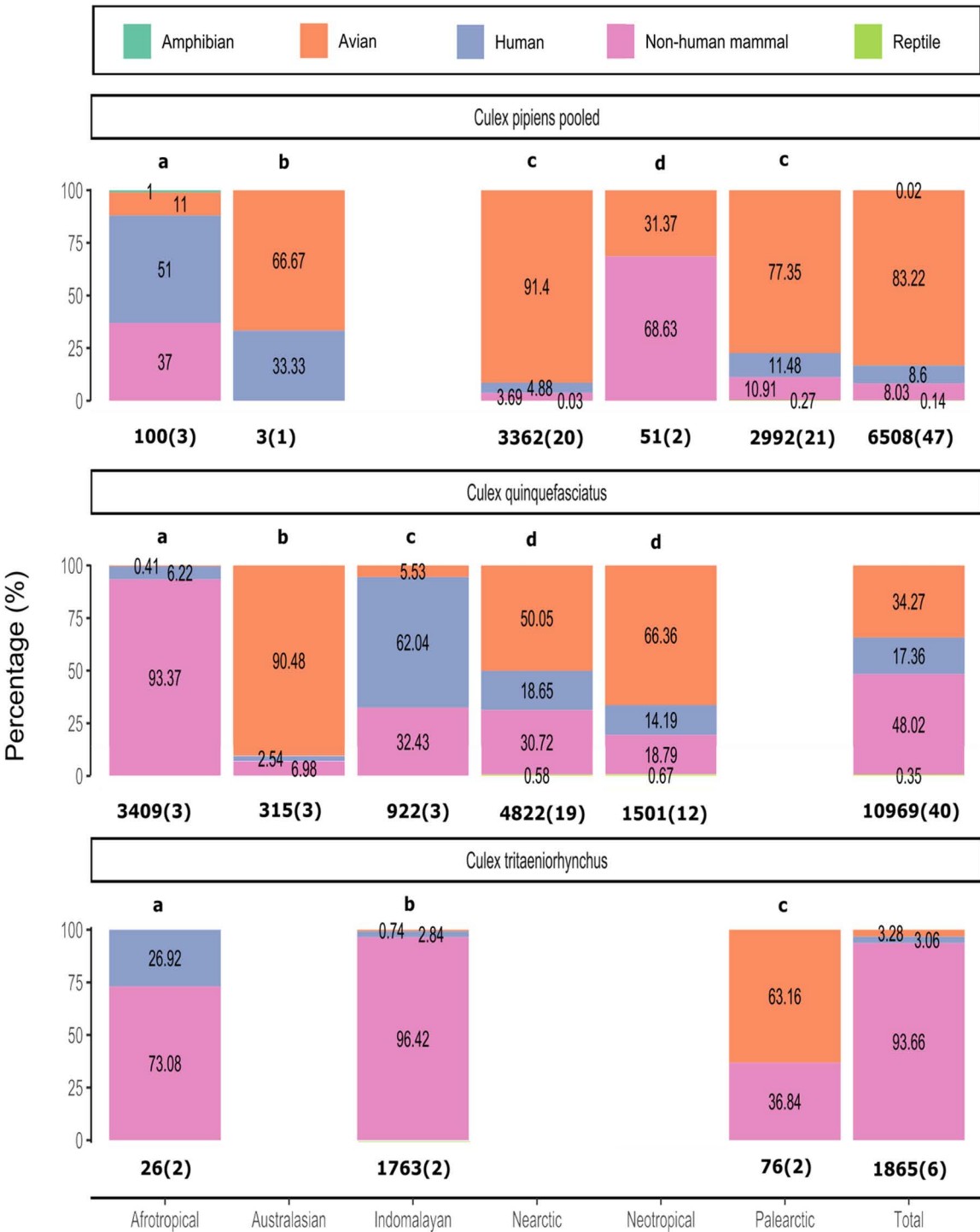

**Fig 3. Feeding patterns of '*Culex pipiens* pooled', *Culex quinquefasciatus*, and *Culex tritaeniorhynchus* in different biogeographical realms.** Percentage of blood-meals on amphibian, avian, human, non-human mammal, and reptile hosts are shown for each realm. The total number of blood-meals and total number of publications (blood-meals(publications)) is shown below each bar. '*Culex pipiens* pooled' consists of the following mosquitoes: *Culex pipiens* not specified (ns), *Culex pipiens molestus*, *Culex pipiens pallens*, *Culex pipiens pipiens*, *Culex pipiens/molestus* hybrid. Bars with the same letter above them do not differ significantly (P > 0.05) in pairwise comparisons by Chi-square. See S3 Table for raw χ2, raw P vales, and corrected P values using the Bonferroni correction method. Pairwise comparisons of realms were undertaken for each species; pairwise comparison of realms between species was not undertaken. The numbers in the bar represent the percentages of feeds on each host group for a specific species in that realm. The variation between studies for this analysis is shown in S8 Fig.

from avian hosts. *Culex quinquefasciatus* is often believed to be a mosquito that can feed on a wide variety of hosts, depending on the abundance of these hosts [37]. This is also similar to the results we found as around 34% of all *Cx. quinquefasciatus* feeds was on avian hosts, 17% on humans, 48% on non-human mammals. Very little research has been done on *Cx. modestus* despite this species being a known highly competent vector for WNV in the lab [38]. Our results show *Cx. modestus* feeds being 97% on avian hosts. Observations from the UK seem to support high avian and mammalian biting patterns [39]. The combination of feeding on both birds and humans emphasizes how this species must not be overlooked as a potential WNV vector. There were several other species which had feeding patterns that included both avian and mammalian hosts but are not well known WNV vectors (*Cx. erythrothorax, Cx. nigripalus, Cx. orbostiensis, Cx. orientalis, Cx. perexiguus, Cx. restrictor, Cx. sitiens*). These species could potentially be implicated in a future outbreak, but perhaps this has not been observed or the species hasn't been in contact with the virus yet. Thus, being aware of the feeding patterns of mosquito species which are currently not well-known vectors is still relevant to assess future risks.

## Comparison of *Culex* phylogeny and feeding patterns

We compared the phylogeny of mosquitoes from the *Culex* genus to their feeding patterns. Hierarchical clustering of *Culex* feeding patterns revealed a different grouping than the clusters of a *Culex* phylogeny based on the *COI* gene. No relationship between *Culex* phylogeny and feeding patterns was observed. In S7 Fig, the feeding patterns are directly lined up with the phylogeny, which also shows no clear relationship between feeding and phylogeny. This is in line with the results from another meta-analysis which summarized the feeding patterns of Australian mosquitoes [40]. However, contrasting results were found in another study looking at the feeding patterns of 256 mosquitoes from multiple genera. Their results indicate that evolutionary relationships and major continental drift events are associated with determining feeding patterns [17]. Our results only focus on one genus and recent feeding patterns, and do not show an association between *Culex* evolutionary relationships and host feeding patterns. By focusing on one genus, there is no influence of distantly related genera creating clusters [17]. Our genus-specific analysis highlights the differences in feeding patterns at a higher resolution, showing that at genus-level analysis, species clusters do not fully explain feeding patterns.

## Feeding patterns of medically important species across realms

For three geographically widespread and important vector species, we analyzed feeding patterns across different biogeographical realms. For '*Cx. pipiens* pooled', *Cx. quinquefasciatus* and *Cx. tritaeniorhynchus* we observed different host utilizations in different realms. This indicates that the presence of the same species can represent a different risk for different areas. For example, whilst *Cx. quinquefasciatus* might pose a limited threat to humans in the Australasian realm, it could be an important bridge vector in the Afrotropical and Nearctic realm. The reason behind these differences in feeding patterns is unknown, but multiple factors such as host availability, land use, and climate could influence this effect.

## Limitations

This difference in feeding patterns across realms could be due to a different host availability. Since we did not have data on the availability of host species in each specific study area, we cannot determine whether a high percentage of feeds on a specific host species was due to an actual preference of the mosquito for that host or if it was due to that host species being the

most available host compared to other species. A more sophisticated way to measure this, if the data were available, would be by utilizing a forage ratio analysis where the observed frequency of blood-meals on a specific host is divided by the relative abundance of that host in the study area [41]. Including an estimate of host availability in future field studies would allow for this analysis of feeding patterns. This could be done using regional host density data where it exists.

Shifts in mosquito feeding behaviour could impact the virus transmission risk to humans. In the US a shift in feeding patterns was observed in *Cx. pipiens* after their main host American robins migrated, which resulted in a switch to humans as their primary host. This increase in *Cx. pipiens* feeding on humans coincided with an increase in people being infected with WNV. A similar pattern was observed for *Cx. tarsalis* where an increased number of human WNV infections occurred if the preferred avian host migrated [42]. Thus, as land use change and climate change could drive mosquitoes to feed on different hosts [9,10], this could alter the potential of virus transmission.

Land use could be an important factor in determining host utilization of these mosquitoes. Geographically distinct populations of mosquitoes can have different host preferences [43]. Urbanization affects feedings patterns as well, with *Cx. pipiens* and *Cx. quinquefasciatus* having increased levels of human feeding in urban areas [10]. Where available we collected the data on the different land use types included in the studies of this meta-analysis (S1 Database). We summarized the feeding patterns of each *Culex* species on the major host groups across different land use types in S2 Fig. Additionally, whether the collection occurs inside or outside can have an effect, with inside collections having a significantly higher percentage of human blood feeding compared to collection that just occurred outside [44]. If specified, the collection location (indoor/outdoor) and the collection method was also included in our database as well (S1 Database). We analyzed the different collection locations and collection method for '*Cx. pipiens* pooled', *Cx. quinquefasciatus, and Cx. tritaeniorhynchus* in S3 and S4 Figs. These figures show that for each of these species relatively similar collection locations and methods were used. Most of the included studies did include some form of description of land use types, however the descriptions did greatly vary in the amount of detail and often lacked a clear quantification of land-use type. This made it challenging to draw any conclusions from the land use data. Future studies should aim to include a description of land use types in a 1 km radius around the collection point of the bloodfed mosquito. They should also give a clear reason why they classify a certain area as urban, rural, or natural. If the data are not available, studies should provide additional information and/or supplementary pictures of the location of the mosquito-collections. Specific information on whether these collections were performed indoor or outdoor should also be given.

When looking at the differences in feeding patterns between realms there was a large variation in the number of blood-meals recorded in each realm. Additionally, the number of publications available for each species varied greatly. We found a moderate correlation between the number of publications and the number of species, so the countries which had more publications likely had higher numbers of *Culex* species that were included (S5 Fig). As we did not correct for sampling effort the results might be biased towards feeding patterns that are common in more well sampled areas. Additionally, for some species complexes that are challenging and inconsistent to identify, such as the *Culex pipiens* complex, data on individual sub-species/biotype are lacking in many included studies. Consequently, our analyses may not accurately reflect the feeding patterns of lower-order taxa, e.g., sub-species for this complex. For most of the studies the local host availability, season, land use, exact time and location of the collected mosquito was not recorded. These details would have given a more comprehensive understanding of the mosquito host feeding patterns. '*Culex pipiens* pooled'

showed variation in feeding patterns between realms. The different subtypes contained in this pool have been shown to have different feeding preferences. *Culex pipiens pipiens* and *Culex pipiens molestus* are an example of this, with biotype *pipiens* having a high percentage of avian blood-meals whilst biotype *molestus* had a higher percentage of human blood-meals [45]. Thus, the differences in feeding patterns between the realms for '*Culex pipiens* pooled' could have been due to the different *Culex pipiens* subtypes contained in the pool being mostly present in different locations.

## Conclusion

Analyzing the feeding patterns of *Culex* mosquitoes on a global scale showed a large variety in feeding patterns between the *Culex* species and a limited genetic basis for these differences. Additionally, variations in feeding patterns in the different realms for the same species were found, highlighting the importance of local surveillance. The reason behind these different patterns is unknown but indicates extrinsic factors such as host availability, land use and climate playing a large role. When determining how host feeding patterns influence human disease risk a wide variety of external factors should therefore be included in the consideration. Finally, as climate and land use change may alter the feeding patterns on avian and human hosts, there may be an increased potential of mosquito-borne viral epidemics and cross species transmission. There are not enough data available to predict this shift and its possible effects at present, again highlighting the importance of local and periodic surveillance in high-risk areas.

## Supporting information

**S1 Database. Full database of *Culex* blood-meal analysis.** The columns of the table give the following details: PMID, title, Authors, First author, Journal/book, Publication year, DOI, Mosquito species, Mosquito location (country), Biogeographical realm, Host (to level described in literature), Host (major host group: amphibian, avian, human, non-human mammalian, and reptilian), Number of blood-meals, Total number of blood-meals (in that specific study), Collection method, Bloodmeal analysis method, Land use type (of where study was done), Indoor/outdoor collections, notes, land use simplified (land use categorized into one of six main categories).
(XLSX)

**S1 Code. Code utilized in this meta-analysis.**
(R)

**S1 Fig. Screening process.** PRISMA flowchart of selection and inclusion process for the meta-analysis of *Culex* blood-meal studies.
(DOCX)

**S2 Fig. Land use types for each *Culex* species.** *Culex* feeding patterns across different land use types. Each figure shows the percentage of blood-meals taken on five major host groups: amphibian, avian, human, non-human mammal, reptile per mosquito species.
(DOCX)

**S3 Fig. Collection methods used for '*Culex pipiens* pooled', *Culex quinquefasciatus, and Culex tritaeniorhynchus.*** Percentage of blood-meals per major host group: amphibian, avian, human, non-human mammal, and reptile, that were collected using different methods, or combinations of methods.
(DOCX)

**S4 Fig. Collection location for '*Culex pipiens* pooled', *Culex quinquefasciatus, and Culex tritaeniorhynchus.*** Percentage of blood-meals collected indoor, mixed (=indoor and outdoor), not specified, and outdoor per major host group: amphibian, avian, human, non-human mammal, and reptile.
(DOCX)

**S5 Fig. Correlation between the number of publications against the number of *Culex* species.**
(DOCX)

**S6 Fig. Variation in number of blood-meals between studies per *Culex* species.** Number of blood-meals taken from each major host group (amphibian, avian, human, non-human mammal, and reptile) for each mosquito species included in this meta-analysis.
(DOCX)

**S7 Fig. *Culex* phylogeny aligned with *Culex* feeding patterns.** *Culex* phylogeny and the percentage of feeding patterns on 5 major host groups (amphibian, avian, human, non-human mammal, reptile) per *Culex* species. The phylogenetic tree is based on the cytochrome c oxidase subunit I (COI) gene of 57 *Culex* (Cx.) mosquitoes.
(DOCX)

**S8 Fig. Variation across studies for the realm analysis.** Feeding patterns of *Culex tritaeniorhynchus* (A), '*Culex pipiens* pooled' (B), and *Culex quinquefasciatus* across different realms. For each realm the total number of blood-meals and total number of publications (blood-meals(publications)) is shown.
(DOCX)

**S1 Table. Overview of data and citations per *Culex* species.** Number of blood-meals and number of publications recorded per *Culex* species. For each species the references describing the blood-meals are shown.
(DOCX)

**S2 Table. Accession numbers of mosquitoes utilized to create phylogenetic tree.** The table indicates for which mosquitoes a full/partial COI gene was available. For those with accessible COI genes an accession number is given.
(XLSX)

**S3 Table. Full statistical analysis of feeding across realms.** The table shows pairwise chi-square comparisons of all included realms per species. Chi-squared values and P values are shown for each comparison. All P values associated with significant differences are indicated in bold.
(DOCX)

## Author contributions

**Conceptualization:** Jet S. Griep, Olena Riabinina, Maya Wardeh, Marcus S. C. Blagrove.

**Data curation:** Jet S. Griep, Eve Grant.

**Formal analysis:** Jet S. Griep.

**Funding acquisition:** Matthew Baylis, Marcus S. C. Blagrove.

**Investigation:** Jet S. Griep, Eve Grant.

**Methodology:** Jet S. Griep, Jack Pilgrim.

**Supervision:** Jack Pilgrim, Olena Riabinina, Matthew Baylis, Marcus S. C. Blagrove.

**Validation:** Jet S. Griep.

**Visualization:** Jet S. Griep.

**Writing – original draft:** Jet S. Griep.

**Writing – review & editing:** Jet S. Griep, Eve Grant, Jack Pilgrim, Olena Riabinina, Matthew Baylis, Maya Wardeh, Marcus S. C. Blagrove.

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
