## [Decision Letter · Decision Letter 0]

19 Aug 2024

Dear Miss Griep,

Thank you very much for submitting your manuscript "Meta-analyses of Culex blood-meals indicates strong regional effect on feeding patterns" for consideration at PLOS Neglected Tropical Diseases. As with all papers reviewed by the journal, your manuscript was reviewed by members of the editorial board and by several independent reviewers. In light of the reviews (below this email), we would like to invite the resubmission of a significantly-revised version that takes into account the reviewers' comments. 

We cannot make any decision about publication until we have seen the revised manuscript and your response to the reviewers' comments. Your revised manuscript is also likely to be sent to reviewers for further evaluation.

Sincerely,

Felix Hol

Academic Editor

Audrey Lenhart

Section Editor

Reviewer's Responses to Questions

**Key Review Criteria Required for Acceptance?**

**Methods**

-Are the objectives of the study clearly articulated with a clear testable hypothesis stated?

-Is the study design appropriate to address the stated objectives?

-Is the population clearly described and appropriate for the hypothesis being tested?

-Is the sample size sufficient to ensure adequate power to address the hypothesis being tested?

-Were correct statistical analysis used to support conclusions?

-Are there concerns about ethical or regulatory requirements being met?

Reviewer #1: The authors selected the COI gene to compare mosquito phylogeny and feeding patterns. While the COI gene is a commonly used gene for species identification, it may not be the best gene for phylogenetic analyses. A recent study used 709 genes for broad-level mosquito phylogeny and analyzed close to 300,000 records of blood meals from 422 species (Soghigian et al. 2023, Nature Communications). Using a similar approach in which multiple genes are used to reconstruct the mosquito phylogeny is more appropriate and this may impact the associations that are found with feeding preferences.

Reviewer #2: - The objectives and hypothesis are clearly stated

- A meta analysis is appropriate for addressing the objectives

- The studies and quantity thereof are appropriate for addressing the questions being asked

- Re: statistical analyses - it's unclear if the large set of pairwise comparisons were corrected for multiple comparisons

- No ethical concerns

Reviewer #3: -The hypothesis of the study is confused because authors mix medical importance in a overall genus study that include species without medical importance. I recomend define two hypothesis: one within the genus analysis and another focus on the medical important species. 

-ref40: it should be in the introduction

-lack of consistency in representing pipiens and molestus forms. Culex quinquefasciatus is part of the Cx. pipiens complex and authors may use Cx. pipiens s.l. and Cx. pipiens s.s. to differenciate or include Cx. quinquefasciatus in the discussion or to represent an agregate analysis within the species or among the subspecies/forms of Cx. pipiens.

-line 48-49: I will remove this sentence because most mosquito genus with vector competence occur across multiple continents and habitats. It is unlikely that these two references claim that Culex genus is special because their species occur in both tropical and temperate regions.

-lines 50-52: Some concepts in this sentence are being miss use. Reservoir host is an important concept for the dynamic of some diseases when the etiological agent can survive in a residual way within this vertebrate host, but it is normally not used for vectors. Bridge vector is a very specific concept for WNV transmission and I should avoid the generalization. 

-The bibliography search require more detail. Which species require further analysis beyond genus search? Which molecular methods were accepted? How many articles were found, how they were defined as valid. I recomend folow some guidelines from PICOS or SPIDER methods to improve the description of the data collection.

-Some statistical analysis are described but not presented. Authors need to improve the application of the methods.

**Results**

-Does the analysis presented match the analysis plan?

-Are the results clearly and completely presented?

-Are the figures (Tables, Images) of sufficient quality for clarity?

Reviewer #1: In Figure 2, the x-axis is missing for the mosquito phylogeny and there is no branch support included. There also appears to be a typo in the legend: “The height of the trees represents the distance between clusters”. Not the height of the tree but the branch length indicates genetic distance.

The lack of more detailed information on collection location, host species, and host abundance prevents the current study from increasing the understanding of the underlying drivers of host-associations beyond describing the associations. Moreover, the lack of data on accurately identified species/subspecies/biotypes, which are known to have different feeding preferences, prevents a more detailed analysis of the broader feeding patterns of these lower-level groups that are important for risk assessments. I appreciate that the authors acknowledge these limitations (and provide recommendations for how to improve future studies), but, unfortunately, this does limit the overall impact of the study.

Reviewer #2: - The analysis leaves out important potential confounders such as collection site type (eg rural/urban/natural) and collection methods, which could be driving the observed results.

- As noted in the main comments, the annotation for Figure 2 needs to be more clearly described

Reviewer #3: -I notice that some articles that are within the criteria described are not included in the Table S1. It may be important have the journal information. I notice that some articles in Parasites& vectors are missing, but it is difficult to check for missing these information.

-Table showing number of article per species should be presented. All articles should be cited. Include number individuals and/or populations per species may be important to verify the robustness of the data per species. 

-Figure 1: Cx. pipiens branch is odd. How do you have phylogeny from COI with Cx. pipiens pooled?; Missing bootstraps information in Tree and other overall statistical analysis results; The combination of the feeding dendrogram and the tree is confuse, and it is unclear it stastitical power of the dendrogram. e.g. Most Culex pipiens s.l. present higher feeds in birds (with similar proportions of mammals) but they are separated in your representation. Representing the proportion chart in the same order than the phylogentic tree will be more useful to discuss differences between close species; I think that you should also represent the variation across studies (or populations) in a plot figure with a median across studies (or populations). This will allow further statistical analysis to verify if the variation observed is statistically different or not (95% confident intervals can be useful for this)

-line 185: it is unclear the goal for this analysis. Why these groups? the table with the number of studies per species is important to justified this selection; The study is justified due their medical importance for arbovirus transmission but them you cluster the information of Culex pipiens s.l. all together; and lack a proper analysis of all vectors; lack of statistical analysis to discuss this topic. You should define a clear goal and perform a depper analysis with key species. 

-line 220: you should provide further analysis in these species... This may be also clear in the introduction. Maybe label species based on their medical importance in the first table suggested.

**Conclusions**

-Are the conclusions supported by the data presented?

-Are the limitations of analysis clearly described?

-Do the authors discuss how these data can be helpful to advance our understanding of the topic under study?

-Is public health relevance addressed?

Reviewer #1: (No Response)

Reviewer #2: - The discussion was well-laid out - specific critiques are included in the comments provided

- Some limitations are described well, others should be addressable in the analyses

- The public health relevance is well addressed

Reviewer #3: -line 268-272: Domestic animals is also a key factor. The preference of mosquito can also be evaluated by the location of the collection. Gomes et al. 2013 found differences between animal shelters in blood fed females. This type of data may also improve your analysis about the topic (especially for the species with medical importance).

-Authors should reorganize the discussion after to improve the definition of the hypothesis and the analysis.

**Editorial and Data Presentation Modifications?**

Reviewer #1: (No Response)

Reviewer #2: (No Response)

Reviewer #3: (No Response)

**Summary and General Comments**

Reviewer #1: Griep et al. performed a meta-analysis of Culex mosquito feeding patterns. The authors analyzed blood feeding data from 90 publications comprising 26,857 blood meals from 71 different Culex species. For all species, mosquito phylogeny was compared with feeding patterns and for a subset of species a more detailed comparison across biogeographical realms was performed. The authors found that feeding patterns were not significantly explained by mosquito species level phylogeny, and they found variation in feeding patterns between realms. The main conclusions of this study are that feeding patterns vary between Culex species and geographically distinct populations. While the study addresses an important question regarding the feeding patterns of Culex mosquitoes, which is relevant to better understand the risk for pathogen transmission, the overall novelty and significance are limited.

Reviewer #2: This manuscript is a meta-analysis of Culex biting patterns over last 15 years, based on studies that performed molecular analysis of blood meals to the species-level. The topic would be of interest to PLOS NTD readers, given that Culex mosquitoes can serve as vectors for diseases, including those with both animal and human hosts. The authors show that there is no apparent association between genetic relatedness of mosquito species and their preferred blood meal hosts. They also show that within 3 individual species, blood meal hosts differ across biogeographic realms. These results are interesting because they suggest that biting preferences cannot be inferred from Culex species alone, but rather they are specific to local environments. The paper is concise and interesting, however I worry that some major confounders may have been overlooked in the analyses and therefore could threaten the conclusions. Thus, major revisions should be required prior to considering for publication in PLOS NTD.

Major concerns:

1. As the authors point out in the discussion, it is well known that the environment surrounding collections (eg indoor/outdoor/urban/rural/natural/etc) can impact the species distribution. It’s hard to imagine that the studies included in these analyses do not provide information about where the collections occurred, and it would be important to know whether the differences in the within-species, across-realm host preferences may be attributable to the environment surrounding collections. Thus, within-species, across-environment comparisons may be valuable for interpreting the findings presented here. Site descriptions are 

2. Additionally, how mosquitoes are collected (eg passive trapping vs aspiration vs etc) can impact the species distributions observed. As with environment, the collection methods for the papers analyzed are not discussed and could be similarly driving some of the differences observed. Collection approaches should be reported in the methods sections of the papers analyzed. 

3. Regarding the above two points: in a quick check on the first 6 papers in Supplemental Table 1, 5 papers described the site types (eg urban, rural, natural) and collection approaches (eg baited traps, aspiration, etc). Thus, this information should be possible to attain and include in analyses.

Other comments:

4. Line 103: What was the justification for including zoos? This choice doesn’t seem like it would be representative of a particular biogeographic realm. How do the results look when papers with zoos are removed?

5. Line 157-158/Figure 1: The number of publications and the number of species per geographic location appear to be quite correlated. Can you please quantify this correlation? This should be added to the discussion (Lines 291-292) about sampling effort.

6. Line 173: Is BGK the same thing as the BGI defined in the methods (line 145)?

7.Lines 196-197: Did you take into account multiple comparisons/df when determining the significance of pairwise comparisons?

8. In general, the notation and legends for Figure 2 need to be clarified and more precise. For example: 

a. Lines 201-202: The number of bloodmeals and publications are below the bars

b. Lines 205-207: Are these superscript letters? Or does this refer to the percentages on the bar plots?

9. Lines 220-222: Do the WNV-competent vectors include avian and human hosts across all biogeographical realms?

10. Lines 229-230: I’m not sure that unpublished results are appropriate for a meta-analysis, but at a minimum, it would be good to have some sort of quantitation on this statement about high human biting behavior of Cx. modestus, and perhaps some speculation on why the UK results are different from the results of your paper.

11. Lines 248-249: You mention here that Cx. tritaeniorhynchus had similar feeding patters across realms, but Fig 2 and Supplemental table 4 indicate that host distributions were different across realms.

12. Lines 273-278: As stated above, most studies likely describe how and where collections occur.

Reviewer #3: The manuscript approuch an interest topic for mosquito biology, but authors should improve how they present the information and define better the goals fo the manuscript.

PLOS authors have the option to publish the peer review history of their article (what does this mean? ). If published, this will include your full peer review and any attached files.

**Do you want your identity to be public for this peer review?** For information about this choice, including consent withdrawal, please see our Privacy Policy .

Reviewer #1: No

Reviewer #2: No

Reviewer #3: Yes: Bruno Gomes

Figure Files:

Data Requirements:

Please note that, as a condition of publication, PLOS' data policy requires that you make available all data used to draw the conclusions outlined in your manuscript. Data must be deposited in an appropriate repository, included within the body of the manuscript, or uploaded as supporting information. This includes all numerical values that were used to generate graphs, histograms etc.. For an example see here: http://www.plosbiology.org/article/info%3Adoi%2F10.1371%2Fjournal.pbio.1001908#s5 .
---

## [Editor Report · Decision Letter 1]

6 Dec 2024

PNTD-D-24-00728R1Meta-analyses of Culex blood-meals indicates strong regional effect on feeding patternsPLOS Neglected Tropical Diseases  Dear Dr. Griep, Thank you for submitting your manuscript to PLOS Neglected Tropical Diseases. After careful consideration, we feel that it has merit but does not fully meet PLOS Neglected Tropical Diseases's publication criteria as it currently stands. Therefore, we invite you to submit a revised version of the manuscript that addresses the points raised during the review process. Please submit your revised manuscript within 30 days Jan 05 2025 11:59PM. If you will need more time than this to complete your revisions, please reply to this message or contact the journal office at plosntds@plos.org. Please include the following items when submitting your revised manuscript: * A rebuttal letter that responds to each point raised by the editor and reviewer(s). You should upload this letter as a separate file labeled 'Response to Reviewers '. This file does not need to include responses to any formatting updates and technical items listed in the 'Journal Requirements' section below. * A marked-up copy of your manuscript that highlights changes made to the original version. You should upload this as a separate file labeled 'Revised Manuscript with Track Changes '. * An unmarked version of your revised paper without tracked changes. You should upload this as a separate file labeled 'Manuscript '. If you would like to make changes to your financial disclosure, competing interests statement, or data availability statement, please make these updates within the submission form at the time of resubmission. Guidelines for resubmitting your figure files are available below the reviewer comments at the end of this letter. We look forward to receiving your revised manuscript.Kind regards, Felix HolAcademic EditorPLOS Neglected Tropical DiseasesAudrey LenhartSection EditorPLOS Neglected Tropical Diseases 

Shaden Kamhawi

co-Editor-in-Chief

Paul Brindley

co-Editor-in-Chief

**Additional Editor Comments:** Many thanks for submitting this revised and improved manuscript. After reading the revisions I would like the below two questions to be clarified before moving on with this manuscript:

1. In response to the reviewer question regarding statistical methods it is stated that "No multiple testing correction was applied". Could the others explain their reasoning behind this choice?

2. molecular methods: what is exactly meant with 'agarose gel diffusion'? Does this mean PCR followed by gel electrophoresis? If yes, agarose gel diffusion is not the conventional way to refer to this. If no, please clarify.**Journal Requirements:**

Please amend your detailed Financial Disclosure statement. This is published with the article. It must therefore be completed in full sentences and contain the exact wording you wish to be published. Please ensure that the funders and grant numbers match between the Financial Disclosure field and the Funding Information tab in your submission form. Note that the funders must be provided in the same order in both places as well.

**Reviewers' comments:** **Figure resubmission:** While revising your submission, please upload your figure files to the Preflight Analysis and Conversion Engine (PACE) digital diagnostic tool, https://pacev2.apexcovantage.com/ . PACE helps ensure that figures meet PLOS requirements. To use PACE, you must first register as a user. Registration is free. Then, login and navigate to the UPLOAD tab, where you will find detailed instructions on how to use the tool. If you encounter any issues or have any questions when using PACE, please email PLOS at figures@plos.org. Please note that Supporting Information files do not need this step. If there are other versions of figure files still present in your submission file inventory at resubmission, please replace them with the PACE-processed versions.**Reproducibility:** To enhance the reproducibility of your results, we recommend that authors of applicable studies deposit laboratory protocols in protocols.io, where a protocol can be assigned its own identifier (DOI) such that it can be cited independently in the future. Additionally, PLOS ONE offers an option to publish peer-reviewed clinical study protocols. Read more information on sharing protocols at https://plos.org/protocols?utm_medium=editorial-email&utm_source=authorletters&utm_campaign=protocols

---

## [Editor Report · Decision Letter 2]

6 Jan 2025

Dear Miss Griep,

We are pleased to inform you that your manuscript 'Meta-analyses of Culex blood-meals indicates strong regional effect on feeding patterns' has been provisionally accepted for publication in PLOS Neglected Tropical Diseases.

Best regards,

Felix Hol

Academic Editor

Audrey Lenhart

Section Editor

Shaden Kamhawi

co-Editor-in-Chief

Paul Brindley

co-Editor-in-Chief

Many thanks for submitting and revising this interesting manuscript. I look forward to seeing it published in PLOS NTD.

all the best,

Felix

---

## [Editor Report · Acceptance letter]

Dear Miss Griep,

We are delighted to inform you that your manuscript, "Meta-analyses of *Culex* blood-meals indicates strong regional effect on feeding patterns," has been formally accepted for publication in PLOS Neglected Tropical Diseases.

Best regards,

Shaden Kamhawi

co-Editor-in-Chief

Paul Brindley

co-Editor-in-Chief
